# Understanding Tourists' Behavioral Intention and Destination Support in Post-pandemic Recovery: The Case of the Vietnamese Domestic Market

**Long Hai Duong [1], Quyet Dinh Phan [2,\*], Tung Thanh Nguyen [3], Da Van Huynh [4] , Thong Tri Truong [3] and Khanh Quoc Duong [5]**

[1]  Department of Global Hospitality and Tourism Management, Kyung Hee University, Seoul 130-701, Korea
[2]  Department of External Affairs and Communication, Thuongmai University, Hanoi 122868, Vietnam
[3]  School of Tourism, Kien Giang College, Rach Gia 920000, Vietnam
[4]  School of Social Sciences and Humanities, Can Tho University, Can Tho 900000, Vietnam
[5]  School of Tourism and Hospitality Management, Royal Roads University, Victoria, BC V9B 5Y2, Canada
\*  Correspondence: quyetphan@tmu.edu.vn

**Abstract:** Many countries have recently strived to accelerate the tourism recovery process by restarting their tourism industry despite the unprecedented risks of the COVID-19 crisis. Noticeably, several tourism destinations have experienced an impressive revitalization of both domestic and international tourist arrivals right after lifting all social distance restrictions. However, little is known about how a tourist destination may revive from the pandemic and to what extent tourists are willing to support a destination recovery. This study, therefore, aims to examine factors influencing the travel demand of domestic tourism and tourists' willingness to support a destination recovery in new normal conditions. The Partial Least Square-Structural Equation Modeling was employed to predict the structural model derived from a sample size of 695 valid questionnaires. The results indicate that there is a significant improvement in domestic tourists' travel intention and their willingness to support the post-pandemic destination revival. It is interesting to learn that the destination health risk image is no longer a critical determinant to tourists' travel plans, while other factors including attitude, monetary promotion, and social media significantly influence their travel intention and support of tourism destination re-opening in new normal conditions. Theoretically, this study generates important contributions to post-disaster crisis management and predicting tourists' behavioral intentions that may influence tourism destination recovery prospects. Practically, the study also provides several important implications to rebuild the domestic tourism industry in a more resilient way against future pandemic challenges.

**Keywords:** post-pandemic recovery; tourism reopening; behavioral intention; destination support

## 1. Introduction

The global health pandemic crisis has triggered huge destruction to the global economy which is projected to suffer an unexpected loss of around USD$ 8 trillion in output by 2023 [1,2]. According to the United Nations [3], global tourism, a vulnerable and slow recovery sector, is likely unable to overcome the "knock-down" hit which is estimated to reach about $ 2 trillion in revenue loss in 2021. Unfortunately, the pandemic crisis may continue threatening the capacity of international destination revival in both short-term and long-term scenarios [4]. More seriously, the tourism-dependent countries seem to be the hardest hit of the pandemic impact and the socio-economic loss, and may take longer to revive after the crisis [5]. Indeed, the lingering lockdowns of the tourism business to stop the spreading of the pandemic has entailed an increasing unemployment rate, business shutdowns, poverty concerns, and social crimes in many countries [1]. As such, all countries must devise proper action plans to adapt to the unprecedented ongoing COVID-19 impact.

Facing the increasing poverty and socio-economic crisis, most countries have made great efforts to reopen their tourism industry despite the existing health-related risks [6]. A wide range of practical measures has been taken into consideration to stimulate the tourism demand and facilitate destination capacity to meet tourists' expectations in post-pandemic [7]. By restarting a part of the tourism industry during the pandemic, a few countries have gained encouraging results thanks to the green zones—where the tourism activities are prioritized but strictly aligned with the epidemic prevention regulations. The global tourism business in early 2022 rebounded to around 61% below 2019 levels, but it is expected to significantly revitalize throughout the rest of 2022 since the global social restrictions may be released in many parts of the world [8]. More specifically, 45 destinations (including 14 in Asia) have removed or eased the COVID-19-related restrictions since early June 2022. Such striking efforts are assumed to boost the global tourism industry in which the domestic tourism in each country could be prioritized for recovery in advance.

The global health crisis has hit the tourism industry in Vietnam hard [9]. The central government has proposed bold measures to rebuild the national tourism system and reduce the pandemic impacts and mass closure of tourism enterprises. Therefore, Vietnam is not an exception in its efforts to encourage national tourism recovery by lifting all the restrictions for domestic and international tourists since March 2022 [10]. Such a combination of international and domestic tourism has fostered a remarkable achievement during the first half of the national tourism year. According to the latest report for the first quarter of 2022, inbound tourism reached 365.3 thousand arrivals, 4.5 times higher than the same period in 2021. Along with the country's socio-economic rejuvenation, the number of domestic tourists in the first 5 months of 2022 also increased by 243% over the same period in 2021, with the total number of domestic tourists up to 48.6 million. The strong and comprehensive recovery of tourism has also boosted the revenue from travel, accommodation, and catering services. Specifically, tourism revenue in the first 5 months of 2022 increased by 34.7% over the same period last year.

Despite the encouraging recovery of tourism destinations around the world, there are critical issues surrounding the extent to which extent tourists are willing to travel in post-pandemic conditions and support destination recovery. Likewise, there remain unanswerable conundrums regarding whether the COVID-19 pandemic is over, and whether tourism destinations can ensure their "immune system" will effectively respond to future pandemic crises in a more resilient way. As such, this study attempts to examine tourists' travel behaviors and intentions in post-pandemic crises. Importantly, this study also aims to explore key factors influencing tourists' destination recovery support in new normal conditions.

## 2. Literature Review

### 2.1. Tourism Destination Re-opening in Post-COVID-19 Pandemic

The global travel industry has become so vulnerable to the lingering damage of the pandemic impact [11]. To alleviate the enormous losses due to the pandemic consequences, several countries have endeavored to rebuild their tourism industry by prioritizing domestic tourism renaissance and gradually removing the entry restrictions to certain international markets. Despite the existing pressures from health-related risks and additional challenges (e.g., rising oil prices, inflation, political disruption), re-opening tourism businesses to alleviate the economic pressure has progressed across the countries in Europe, America, and Asia [8]. In other words, many countries have redirected their zero-COVID-19 policy to a more adaptive approach to live with the epidemic crisis instead [12]. As a result, the unexpected recovery of the international tourist arrivals during the first quarter of 2022 has reached 117 million compared to 41 million in the same period in 2021 [8]. Travel restrictions have been lifted for international tourists, while the widespread vaccination rollout also strengthened the confidence of both tourism enterprises and tourists to make travel decisions.

Despite the encouraging achievements of the tourism industry in post-pandemic recovery, most tourism destinations are very cautious with crisis management. Governments retain an eye on the epidemic control while a wide range of adaptive measures have been implemented. At an organizational level, the hotel industry has adopted technological approaches to reduce human interaction, while hotel guests are likely to preferably use more smart applications instead. At the destination level, disruptive technological applications (e.g., travel apps) are also developed to help tourists to tailor a safe travel plan while visiting a tourist attraction. With great efforts in a committed way, tourism destinations are likely to be able to foster tourists' travel intention and support tourism revitalization in the post-pandemic crisis.

*2.2. Domestic Tourism and Revival Prospect in New Normal Conditions*

In response to the global mobility restrictions, a lot of countries have fostered domestic tourism as a potential driving force for tourism recovery during and post-pandemic crises. Instead of closing the tourism industry thanks to the social and mobile restrictions, domestic tourism has gradually been reopened to save local businesses [13]. This also means that tourist destination reopening has to be reconciled with the potential risks under the ongoing pandemic prevention pressures [14]. However, tourism experts argue that there will hardly be the so-called "back to normal" as before the global pandemic outbreak, since the tourism transformation to adapt to the current challenges may take years to develop [15].

Despite the difficulties, it is essential for tourism destinations to restructure their tourism businesses in a more resilient way [16]. A requisite advantage to restarting tourism activities is the widespread vaccination rate that inspires not only tourism destination managers but also tourists to co-construct a more adaptive tourism system. Falk et al. believe that the high domestic tourism demand plays a key part to support tourism reopening and recovery in the short term [17]. Several studies have indicated that there are feasible measures to restore the domestic tourists' travel motives in a more sustainable way. Safety-related conditions at tourism destinations are likely influential to tourists' travel decisions. Jeon et al. [18] argue that there would be an alternative to the zero-COVID-19 policy by facilitating domestic tourism services and rebuilding a safe tourism image to push the domestic market. Despite the potential to revive domestic tourism, destinations should take into account proper measures to minimize risks facing tourists while traveling to a destination [19]. Therefore, low-risk destinations such as nature-based attractions are likely more attractive to domestic tourists [20]. Many studies have highlighted the diverse range of tourists' motivations to travel in the post-pandemic period, which contributes to the prospect of tourism rejuvenation in the short-term [21].

In response to the ongoing pandemic impact, the regeneration of domestic tourism could contribute to tourism destination revitalization. First, green zones could be employed as a temporary approach to meet the increasing demand for the domestic market [22]. A few countries have started to promote tourism businesses at green zones which are also known as safe tourism destinations with few or no COVID-19 cases [7]. According to a study by Da et al. [23], the domestic market recovery in the post-pandemic crisis could be feasible if the safe destinations were ensured a unique mechanism. More importantly, the encouraging recovery of a tourism destination might inspire other tourism destinations to gradually rebuild the tourism system [22,24,25]. Lessons learned from successful destinations allow the tourism systems in a country to be regenerated in a way that could better adapt to new challenges [26]. More importantly, a long-term plan to gear up for the international tourism recovery would be essential when more countries lift the border closure and entry restrictions. This also means that a more proactive preparation may entail a more resilient tourism renaissance. Hence, such transformative measures as technological innovation, promotional strategies, and tourism infrastructure should be integrated in such a way that all stakeholders could collaborate effectively to rebuild the tourism businesses [27]. By doing so, tourism scholars believe that domestic tourism recovery is achievable in the

post-pandemic crisis, which provides a strong premise for the national tourism recovery in each country [28].

### 2.3. Effects on Attitude, Perceived Behavioral Control, and Subjective Norm on Tourists' Travel Intention in the Post-Pandemic Period

This study adopts the theory of planned behavior (TPB) as a cornerstone for the theoretical framework development. From the perspective of social science research, TPB has been widely used to understand psychological aspects of human behavior [29]. Essentially, TPB is an extension of a widely accepted theoretical framework known as the theory of reasoned action (TRA) which is employed to understand and predict behaviors [30,31]. TRA assumes that there are causal relationships among key components including common beliefs, attitudes, and subjective norms which could result in certain behavioral intentions [32]. However, this conceptual model might be limited in predicting human behaviors in all situations [32,33]. As such, the extension of TPB is expected to overcome the limitations of the TRA model.

Attitude toward a behavior is one of the pivotal determinants used for explaining and predicting behavioral intention. Ajzen ([30], p. 187) defined attitude as "the perceived ease or difficulty of performing the behavior, and it is assumed to reflect past experiences as well as anticipated impediments and obstacles" [30]. Subjective norms are based on normative beliefs that may lead to perceived social pressure [34]. In other words, subjective norms reflect the personal perceptions about the impact of other people's behavioral performances which may be significantly associated with their behavioral beliefs and intention [35].

The influence of "attitude", "subjective norm", and "perceived behavioral control" on "behavioral intention" varied across contexts [32]. In the particular context of post-COVID-19 recovery, researchers suggest that there is a critical correlation between attitude and tourists' plans to visit a destination during a pandemic outbreak [36–38]. Yang-Wallentin [39] argued that there is a profound impact of tourists' perceived behavioral control on their holiday intention. Under the lingering impact of the COVID-19 pandemic, visitors are more likely to plan their trip based on the recommendations from important referents. Therefore, there may be some critical changes in tourists' perceptions about travel needs in new normal conditions. Although there is limited research examining the effect of attitudes, subject norms, and perceived behavioral control on behavioral intention in the context of post-pandemic destination reopening, scholars suggest that these factors indicate critical correlations [40,41]. Hence, the following assumptions are made:

**Hypothesis 1 (H1).** *Attitude positively affects tourists' travel intention.*

**Hypothesis 2 (H2).** *Perceived behavioral control positively influences tourists' travel intention.*

**Hypothesis 3 (H3).** *Subjective norm positively influences tourists' travel intention.*

### 2.4. Effects of Social Media on Travel Intention, Destination Health Risk Image, and Support for Destination Recovery

Interactive technologies (i.e., all forms of social media) have been recently used as a powerful tool to shape tourists' behaviors [42]. Social media may include different genres of social platforms which could be adopted by destination managers to predict tourism demand [43]. Previous studies suggest that perceived information from social media might be associated with tourists' plans to visit a destination after the COVID-19 pandemic [44–46]. Apart from conventional media (e.g., mainstream television channels), alternative media have increasingly attracted a wide range of audiences' engagement in a more involved way. As such, social media platforms have been used to shape tourists' behavioral intentions and market destination images [47]. Jaya et al. found that social influencers on social media might have a remarkable influence on their followers' travel decisions based on the perceived information about tourism destination im-

ages [48]. This also means that the reliable information provision from media, whether mainstream or alternative media, is more critical in tourists' decision-making process. Aaron et al. [49] found that tourists were likely to seek travel information from multiple social media channels in which word-of-mouth information was even more influential for their travel intentions.

A handful of previous studies also suggest that a safe destination health risk image is likely to influence tourists' willingness to make travel decisions [50,51]. This is quite understandable under the circumstances of the COVID-19 pandemic when health-related risk is probably the biggest concern among travelers [52]. Despite the current improvement in epidemic control, the epidemic is still complicated and unpredictable, re-emerging in some countries with the appearance of the latest variants of the Omicron strain [53]. In order to rebuild a better destination health risk image, scholars suggest that social media could play a critical role in shaping tourists' behavioral intentions. It is also interesting to note that the recovery of domestic tourism in many countries implies that the domestic market demand has gradually recovered [41]. Despite the financial difficulty after the pandemic, more and more tourists are willing to go on holiday [42]. However, there is limited understanding of the extent to which tourists are willing to support a destination rejuvenation after the pandemic shock. Hence, the following hypotheses are proposed:

**Hypothesis 4 (H4).** *Social media positively influences tourists' travel intention.*

**Hypothesis 5 (H5).** *Social media positively influences destination health risk image.*

**Hypothesis 6 (H6).** *Social media positively influences tourists' willingness to support a destination recovery.*

*2.5. Monetary Promotion (MP)*

Monetary promotion in tourism could be employed as a powerful marketing tool to promote tourism demand and enhance destination competitiveness [54,55]. In a particular context, monetary promotion could be used as a temporary measure to foster the travel demand of tourists and enhance the destination image by providing value-added promotions such as service price reduction, holiday package deals, free entrance tickets, and so on. As such, the monetary promotion measure, if done well, could affect tourism consumption, profitability, and growth recovery. However, this pricing strategy may not work in all contexts since it may degrade the destination brand equity in the long term [56]. Prior research has stressed the dominant role of monetary promotion, which might stimulate tourists' willingness to consume more services and products [57,58]. Chua et al. [59] believe there are critical relationships between pricing strategy and tourist travel demand. Under the pressure of the COVID-19 pandemic's lingering economic impact, monetary promotion is likely a more urgent measure to revive tourists' travel needs. In other words, macro-financial assistance should be urgently implemented to support the entire tourism system recovery. From the perspective of tourism governance, the MP strategy could be deployed as an effective tool for the governments to intervene in the financial crisis facing tourism entrepreneurs in the post-pandemic recovery. As such, a plethora of countries have offered special tourism policy measures (e.g., subsidies, tax cuts, or loans) to facilitate the tourism system re-opening. This allows the enterprises to adjust prices in such a way that may directly stimulate domestic demand because of the fact that tourists may face the common status of low or no income after the pandemic [60]. Likewise, tourism enterprises can reinvest in their tourism business and tailor new tourism products and services which better meet the market demand in new normal conditions.

However, price deduction is not necessarily a powerful measure that determines tourists' behavioral intention but a high-quality tourism experience. Recent studies indicate that low-income tourists are willing to pay higher prices for unique tourism packages [7,23]. Ahmad et al. [61] argue that monetary promotion strategy should be flexibly adjusted to suit

different genres of tourist markets. By doing so, it is assumed that monetary promotion may stimulate tourism destination recovery in the short-term and foster tourists' willingness to support a destination. Therefore, this study proposes the following hypotheses:

**Hypothesis 7 (H7).** *Monetary promotion positively influences tourists' travel intention.*

**Hypothesis 8 (H8).** *Monetary promotion positively influences tourists' willingness to support a destination recovery.*

*2.6. Destination Health Risk Image (DHRI), Travel Intention (TI), and Willingness to Support a Destination (WSD)*

Although several countries have recently attempted to reopen their domestic and international tourism businesses, there exist important concerns regarding how to ensure a safe destination image in the long term [52]. During the pandemic, the "travel bubble or green zone" initiative to promote tourism activities between safe destinations has been implemented in many countries. Despite the success or failure, rebuilding a strong destination health risk image not only fosters a more resilient destination but also motivates tourists' travel demand. Song et al. [62] suggest that risk-related factors exposed to tourism destinations may influence tourists' decision-making process. Zhang and Blassco [63] argue that post-crisis management should focus on reconstructing a safe destination image by engaging tourists in the recovery process. By doing so, the authors believe that tourists will support destination recovery in a more involved way. Recent research indicates that tourists are likely to be more responsible and supportive of the safe destination reopening after the pandemic crisis [64]. Low-risk tourism destination image is found to have direct effects on tourists' destination choices. Although the latest reports suggest that current Omicron variants are less severe than the initial ones, tourists remain cautious about the perceived probability of health-related risks at tourism destinations. Ong et al. [65] assert that an unsafe destination image may hinder tourists' travel decisions, which may directly affect the recovery of tourist attractions. Therefore, destination image should be rebuilt in a way that fosters the temporary recovery of domestic tourism and endeavors to transform a more adaptive destination resilience prospectively. According to the above theoretical review, the following hypotheses are suggested:

**Hypothesis 9 (H9).** *Destination health risk image positively influences tourists' travel intention.*

**Hypothesis 10 (H10).** *Destination health risk image positively influences tourists' willingness to support a destination recovery.*

**Hypothesis 11 (H11).** *Tourists' travel intention has a positive influence on tourists' willingness to support a destination recovery.*

## 3. Research Methodology

*3.1. Measurement Instruments*

Since the scope of the study involves exploring the domestic tourists' behavioral intention and willingness to support destination recovery in the post-pandemic crisis, a quantitative approach was employed to shed light on the research phenomenon. Due to social distancing restrictions, an online questionnaire survey was utilized to gather data. To recruit relevant participants for the study, the potential respondents who intended to travel in the post-pandemic crisis were approached by publishing the online survey on common tourism platforms such as tourism agents, tourism associations, travel blogs, tourism boards, and other social networks. The respondents were invited to voluntarily partake in the online survey whilst the research information related to the research project's aims and informants' right to withdraw any provided information were also clarified.

The questionnaire contents were developed based on the proposed theoretical framework which were adjusted to suit the research context (see Figure 1). The questionnaire was initially designed in English language and then was translated into Vietnamese since the vast majority of the participants were domestic tourists. After being reviewed by tourism experts, the modified questionnaire was prepared for a pilot study to test the feasibility of the research approach. The questionnaire design consisted of three parts. The first section aimed to facilitate tourists answering the questionnaire by asking for basic information related to their motivations to travel in the post-pandemic, followed by the second section to collect key information related to the research objectives while the last section covered demographic information of the respondents.

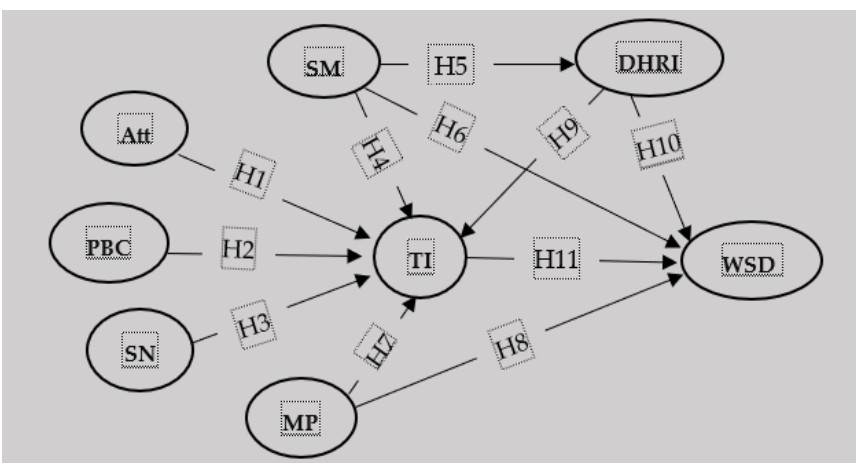

**Figure 1.** Proposed research framework.

Regarding the measurement rationales, this study thoroughly reviewed the literature and adopted relevant theoretical frameworks from previous studies. This study has carefully selected key theoretical components that suit the research context (i.e., the post-pandemic tourism re-opening in Vietnam) and the scope of the study. More specifically, the first construct Social Media (SM) was measured with four items which were adapted from Ebrahimi et al. [66]. Regarding the measurement of the Subjective Norm (SN) construct, four items were selected from a study by Ukpabi et al. [67]. To test the constructs related to Attitude (Att) and Perceived Behavioral Control (PBC), three items were selected for each construct based on the theoretical framework suggested by Sánchez-Cañizares et al. [68]. Referring to the assessment for the construct Destination Health Risk Image (DHRI), four items were adapted from previous literature [68,69]. The construct Monetary Promotion (MP) adapted three items proposed by Chua et al. [59]. Travel Intention (TI) was assessed by three items which were carefully selected from prior studies and were adjusted to suit the research context [69,70]. Ultimately, the four items were adapted from Rastegar et al. to test the factor related to Willingness to Support Destination (WSD) [69]. This study employed a 5-point Likert-type scale ranging from 1 (strongly disagree) to 5 (strongly agree). Accordingly, all of the measurement constructs and items were adjusted to meet the research objectives and the research context under investigation.

### 3.2. Data Collection and Data Analysis

Due to the social distancing regulations associated with COVID-19 in late 2021 in Vietnam, this study used the convenience sampling method to collect data online. The online survey progressed from November 2021 to February 2022 before the entire re-opening of the tourism systems in Vietnam was officially announced on the 15th of March 2022. The population of this study mainly comprised the domestic respondents who had the travel intention in post-pandemic destination re-opening. According to MacCallum et al. [71], the sample size of a study is recommended to be five times larger than the number of

variables to make sure the sample is adequate for estimation. Therefore, adopting a total of 28 measurement items, this study had to collect at least 140 qualified questionnaires to ensure representativeness for the research population. A total number of valid 712 questionnaires were obtained but only 695 qualified questionnaires were utilized for further analysis, which followed the screening process to eliminate disqualified questionnaires (e.g., missing data, unengaged answers, or responses with extreme multivariate- outliners).

Due to the complexity of the research model, the structural equation modeling (SEM) was conducted to test the hypotheses. According to Fornell and Bookstein [72], the least square-based SEM (PLS-SEM) was suitable for testing developmental theories. Accordingly, this study employed PLS-SEM to analyze the data with the Smart PLS 3.3 version. The analysis has systematically assessed the demographic information, measurement model, and structural model.

## 4. Research Results

### 4.1. Respondents' Demographic Profile

This study collected a wide range of relevant information from the respondents including gender, age, educational levels, marital status, and income (Table 1). Regarding the gender difference in this study, the number of female participants (67.3%) outnumbered that of the male informants (32.7%). The majority of the participants were aged from 30 to 45 years old, 46 to 60 years old, and below 30 years old with 36%, 26.3%, and 25.2%, respectively, while the other group aged over 60 made up the minority of the sample size with 12.5%. Referring to the marriage status, the number of single respondents made up 64.9%, as compared with married respondents (34.2%) and divorced informants (0.8%). Concerning the educational levels, the majority of the participants earned at least a bachelor's degree (40.3%) and a graduate degree (34.2%) while the number of respondents with vocational training, high school, and under high school education constituted 20.6%, 2.6%, and 2.3%, respectively. Noticeably, the majority of the respondents generally facing financial difficulties with low and no income made up 67.9%, whereas 20.3% of the participants earned USD$ 474–862 per month. Likewise, the rest of the respondents, who could maintain their monthly income above $ 905 amidst the COVID-19 pandemic, accounted for just 11.8%.

**Table 1.** Descriptive statistics of the respondents.

| Group | Frequency | (%) | Group | Frequency | (%) |
|---|---|---|---|---|---|
| Gender | | | Education | | |
| Male | 227 | 32.7 | Under high school | 16 | 2.3 |
| Female | 468 | 67.3 | High school | 18 | 2.6 |
| Age | | | Vocational high school | 143 | 20.6 |
| Below 30 | 175 | 25.2 | Undergraduate | 280 | 40.3 |
| 30–45 | 250 | 36.0 | Graduate | 238 | 34.2 |
| 46–60 | 183 | 26.3 | Monthly Income | | |
| Over 60 | 87 | 12.5 | No income | 229 | 32.9 |
| Marital Status | | | Below $431 | 243 | 35.0 |
| Married | 238 | 34.2 | $474–862 | 141 | 20.3 |
| Single | 451 | 64.9 | $905–1293 | 45 | 6.5 |
| Divorced | 6 | 0.8 | $1336–1724 | 33 | 4.7 |
| Note(s): $ USA dollars | | | Above $1724 | 4 | 0.6 |

### 4.2. Measurement Model

Assessing the outer model (measurement model) is an important stage of PLS-SEM analysis to predict the relationships between each latent construct and the observed indicator [73]. The measurement model was tested with seven key factors including a total number of 28 measurement items. As shown in Table 2, the results comprised the assessment of construct reliability, validity, and discriminant validity.

**Table 2.** Assessment of the first-order factor model.

| Measurement scales | M | SD | Loadings | CR | AVE |
|---|---|---|---|---|---|
| **Social Media (SM)** | | | | | |
| SM1_In the current situation, photos of this tourism destination on Facebook/ Instagram made me interested to travel | 3.81 | 0.786 | 0.875 | | |
| SM2_In the current situation, the attractiveness of tours to this tourism destination shared on Facebook/ Instagram made me interested to travel | 3.53 | 0.856 | 0.809 | 0.910 | 0.716 |
| SM3_In the current situation, memories that people of this tourism destination shared on Facebook/ Instagram made me interested to travel | 3.80 | 0.826 | 0.863 | | |
| SM4_In the current situation, comments on Facebook/Instagram posts encouraged me to travel to this tourism destination | 3.65 | 0.820 | 0.837 | | |
| **Monetary Promotion (MP)** | | | | | |
| MP1_Tourism-related businesses (e.g., airlines, hotels, restaurants) should offer price discounts when the COVID-19 outbreak is both under control and its adverse impact is minimal. | 4.12 | 0.680 | 0.944 | | |
| MP2_Tourism-related businesses should use price discounts when the COVID-19 outbreak is both under control and its adverse impact is minimal. | 4.16 | 0.685 | 0.949 | 0.947 | 0.857 |
| MP3_Tourism-related businesses should use price discounts more frequently than non-hospitality businesses when the COVID-19 outbreak is both under control and its adverse impact is minimal | 4.00 | 0.787 | 0.883 | | |
| **Subjective norm (SN)** | | | | | |
| SN1_Most people who are important to me think that I should travel | 2.79 | 0.97 | 0.915 | | |
| SN2_Most people whose opinion I value agree with me about traveling | 2.89 | 1.00 | 0.938 | 0.963 | 0.685 |
| SN3_Most people whose opinion I value the support that I travel | 2.77 | 0.99 | 0.955 | | |
| SN4_Most of the people whose opinions I value recommend that I travel | 2.79 | 1.01 | 0.913 | | |
| **Perceived behavioral control (PBC)** | | | | | |
| PBC1_When traveling in Vietnam for now, my health is under my control) | 3.85 | 0.72 | 0.885 | | |
| PBC2_When traveling in Vietnam for now, it is easy to conduct health-protecting activities) | 3.95 | 0.69 | 0.915 | 0.901 | 0.752 |
| PBC3_I have resources, time, and opportunity to protect my health when traveling in Vietnam for now | 4.11 | 0.69 | 0.798 | | |
| **Attitude (Att)** | | | | | |
| Att 1_I think it is good to travel when the epidemic is over | 3.24 | 0.907 | 0.881 | | |
| Att 2_I think it is valuable to travel when the epidemic is over) | 3.23 | 0.903 | 0.919 | 0.944 | 0.809 |
| Att 3_I think it is interesting to travel when the epidemic is over | 3.44 | 0.907 | 0.908 | | |
| Att 4_I think it is delightful to travel when the epidemic is over | 3.38 | 0.916 | 0.890 | | |
| **Destination Health Risk Image (DHRI)** | | | | | |
| DHRI1_I am worried that the accommodation facilities will not be sanitary | 3.49 | 0.926 | 0.804 | | |
| DHRI2_I'm worried that the diet will be unhealthy | 3.37 | 0.945 | 0.772 | 0.895 | 0.680 |
| DHRI3_I'm worried about getting sick during my travel | 3.88 | 0.863 | 0.878 | | |
| DHRI4_I'm afraid that I can't get timely treatment for illness or other physical harm during my travel | 3.64 | 0.983 | 0.842 | | |
| **Travel Intention (TI)** | | | | | |
| TI1_If given the opportunity, I am willing to travel to destination A after COVID-19 | 3.78 | 0.857 | 0.882 | | |
| TI2_I am planning to travel to destination A after COVID-19 shortly | 3.57 | 0.913 | 0.897 | 0.926 | 0.806 |
| TI3_The likelihood of my travel to destination A is high | 3.38 | 1.016 | 0.913 | | |
| **Willingness to Support a Destination (WSD)** | | | | | |
| WSD1_I would encourage my friends and relatives to travel to destination A after the COVID-19 crisis | 3.48 | 0.870 | 0.836 | | |
| WSD2_I say good things about destination A on social media | 3.83 | 0.780 | 0.921 | 0.913 | 0.779 |
| WSD3_I would promote this destination to help tourism recovery | 3.88 | 0.794 | 0.889 | | |

According to Leguina [74], the reliability of a construct requires a high correlation among the indicators to ensure the consistency of measurement items. The internal reliability and validity were tested by composite reliability which was highly recommended to range from 0.6 to 0.7. Table 2 indicated that all the CR values were greater than the 0.7 level, which was highly recommended for the reliability measurement [75].

The convergent validity was also tested to ensure the positive correlation among the items of the constructs. Hair et al. [75] suggested that average variance extracted (AVE) and item loadings could be used to evaluate the convergent validity. As seen in Table 2, all of the Cronbach alpha values were greater than 0.7, which indicates that the construct reliability is valid for further analysis. Likewise, all of the AVE values were greater than 0.5, which also means that all of the constructs used in this study were acceptable since they could explain more than 50% of the variance of the indicators.

Furthermore, the discriminant validity was appraised to check the uniqueness of the constructs presented in the model by considering the cross-loading of indicators, Fornell and Larcker criterion, and Heterotrait–Monotrait ratio [75]. The first discriminant validity evaluation indicated that the factor loading indicators of each assigned construct were higher than that of other constructs [75]. Moreover, the Cronbach alpha CR and Rho_A, as shown in Table 3, are all greater than 0.7, which is commonly accepted for social science research [73,76,77]. Importantly, the convergent validity of the measurement model in this study has met the minimum requirement since all AVE coefficient values are greater than 0.50. Eventually, this study also used Heterotrait–Monotrait ratio to evaluate the discriminant validity and all the values between 0 and 1 indicated that the validity requirement has been obtained.

**Table 3.** Reliability, Validity, and Correlation.

| Variable | Fornell–Larcker Criterion | | | | | | | | Heterotrait–Monotrait ratio | | | | | | | |
|---|---|---|---|---|---|---|---|---|---|---|---|---|---|---|---|---|
| | 1 | 2 | 3 | 4 | 5 | 6 | 7 | 8 | 1 | 2 | 3 | 4 | 5 | 6 | 7 | 8 |
| Att (1) | 0.899 | | | | | | | | | | | | | | | |
| DHRI (2) | −0.035 | 0.825 | | | | | | | 0.039 | | | | | | | |
| MP (3) | 0.220 | 0.168 | 0.926 | | | | | | 0.240 | 0.174 | | | | | | |
| PBC (4) | 0.389 | 0.071 | 0.360 | 0.867 | | | | | 0.435 | 0.098 | 0.411 | | | | | |
| SM (5) | 0.282 | 0.127 | 0.414 | 0.273 | 0.846 | | | | 0.312 | 0.134 | 0.459 | 0.318 | | | | |
| SN (6) | 0.579 | −0.172 | 0.105 | 0.261 | 0.209 | 0.930 | | | 0.620 | 0.172 | 0.112 | 0.286 | 0.234 | | | |
| TI (7) | 0.529 | −0.078 | 0.302 | 0.441 | 0.366 | 0.445 | 0.898 | | 0.585 | 0.079 | 0.334 | 0.508 | 0.415 | 0.487 | | |
| WSD (8) | 0.417 | 0.017 | 0.385 | 0.429 | 0.477 | 0.331 | 0.523 | 0.883 | 0.466 | 0.085 | 0.435 | 0.508 | 0.549 | 0.362 | 0.598 | |
| $\alpha$ | 0.921 | 0.853 | 0.916 | 0.835 | 0.868 | 0.948 | 0.879 | 0.857 | | | | | | | | |
| Rho_A | 0.921 | 0.923 | 0.923 | 0.854 | 0.879 | 0.950 | 0.881 | 0.858 | | | | | | | | |
| CR | 0.944 | 0.895 | 9.47 | 0.901 | 0.910 | 0.963 | 0.926 | 0.913 | | | | | | | | |
| AVE | 0.809 | 0.680 | 0.857 | 0.752 | 0.716 | 0.865 | 0.806 | 0.779 | | | | | | | | |

Notes: $\alpha$ (Cronbach's Alpha), Rho_A (Reliability Coefficient), CR (Composite Reliability), AVE (Average Variance Extract).

### 4.3. Structural Model

This section depicts the status of the structural model suitability and hypothesis testing results (Table 4). According to Parwoll et al. [78], the model fit values have to meet the acceptable levels; this also means that the standardized root means square residual (SRMR) has to be less than 0.08 while the INF value is required to be above 0.80. The SRMR value (0.051) and the NIF value (0.859) confirm that model fit values are at acceptable levels.

Among the eleven hypotheses examined in this study, only one hypothesis was rejected while the rest were supported. Turning to the relationship between "Attitude", "perceived behavioral control", "Subjective norm", and "Travel intention", tourists' travel intention is positively influenced by their attitude ($\beta_{Att \rightarrow TI}$ = + 0.277, t = 6.035, $p < 0.001$). Likewise, perceived behavioral control was found to affect tourists' intention to travel in the post-pandemic period ($\beta_{PBC \rightarrow TI}$ = + 0.218, t = 5.458, $p < 0.001$), while subjective norm positively affected travel intention of the respondents ($\beta_{SN \rightarrow TI}$ = + 0.168, t = 4.345, $p < 0.001$). Thus, the results generally indicate that Hypotheses 1, 2, and 3 were all supported.

**Table 4.** Results of direct effects.

| Hypothesis | Path Coefficient | T Statistics | *p*-Values | Supported/Rejected |
|---|---|---|---|---|
| H1: Att $\rightarrow$ TI | 0.277 | 6.035 | 0.000 *** | Supported |
| H2: PBC $\rightarrow$ TI | 0.218 | 5.458 | 0.000 *** | Supported |
| H3: SN $\rightarrow$ TI | 0.168 | 4.345 | 0.000 *** | Supported |
| H4: SM $\rightarrow$ TI | 0.168 | 4.147 | 0.000 *** | Supported |
| H5: SM $\rightarrow$ DHRI | 0.127 | 2.755 | 0.006 ** | Supported |
| H6: SM $\rightarrow$ WSD | 0.276 | 7.379 | 0.000 *** | Supported |
| H7: MP $\rightarrow$ TI | 0.091 | 2.440 | 0.015 * | Supported |
| H8: MP $\rightarrow$ WSD | 0.161 | 4.132 | 0.000 *** | Supported |
| H9: DHRI $\rightarrow$ TI | −0.092 | 3.047 | 0.002 ** | Supported |
| H10: DHRI $\rightarrow$ WSD | −0.016 | 0.476 | 0.634 | Rejected |
| H11: TI $\rightarrow$ WSD | 0.372 | 9.737 | 0.000 *** | Supported |

Notes: Significant at * $p < 0.05$, ** $p < 0.01$, *** $p < 0.001$.

Turning to the relationship between "Social media", "Destination health risk image", and "Travel intention", it is also worth highlighting that social media has a positive influence on tourists' intention to travel during the post-pandemic recovery ($\beta_{SM \rightarrow TI}$ = + 0.168, t = 4.147, $p < 0.001$), while social media was also found to have a positive effect on destination health risk image ($\beta_{SM \rightarrow DHRI}$ = + 0.127, t = 2.755, $p < 0.01$). Noticeably, social media is a significant determinant of willingness to support a destination ($\beta_{SM \rightarrow WSD}$ = +0.276, t = 7.379, $p < 0.001$). Generally, Hypotheses 4, 5, and 6 were supported.

With reference to the relationship between "Monetary promotion", "Travel intention", and "Willingness to support a destination", the results highlighted a positive correlation between monetary promotion and tourists' travel intention ($\beta_{MP \rightarrow TI}$ = + 0.091, t = 2.440, $p < 0.001$) while financial measures were found to positively affect tourists' support towards tourism destinations during the post-pandemic recovery ($\beta_{MP \rightarrow WSD}$ = + 0.161, t = 4.132, $p < 0.05$). Hence, Hypotheses 7 and 8 were supported.

Despite the importance found in previous studies [79,80], the results in this study showed that there was a negative effect of destination health risk image on travel intention ($\beta_{DHRI \rightarrow TI}$ = −0.092, t = 2.755, $p < 0.01$). Therefore, the 9th hypothesis is supported. However, the results indicated that destination health risk image decreased tourists' willingness to support destination recovery ($\beta_{DHRI \rightarrow WSD}$ = −0.016, t = 9.737, $p > 0.5$). This also means hypothesis 10 was rejected.

Finally, the study also examined the relationship between "Travel intention" and "Willingness to support a destination". The results generally indicated that travel intention positively affected tourists' willingness to support a destination recovery in the post-COVID-19 pandemic ($\beta_{TI \rightarrow WSD}$ = + 0.372, t = 9.37, $p < 0.001$), which also implies that hypothesis 11 is supported in this study.

## 5. Discussion and Conclusions

The connection between the perceived behavioral control, attitude, subjective norm, and intention to travel amidst the COVID-19 pandemic has gained extensive attention among tourism scholars [59,66,68,70,81,82]. However, almost no research examines how monetary promotion, social media, and destination health image influence tourists' travel intention and support toward post-pandemic destination recovery. Given the direct effects of monetary promotion and social media on travel intention, it is argued that financial measures to stimulate domestic tourism are crucial to the recovery of tourism destinations. Previous studies stressed destination health risk image as a determinant for travel intention but the low correlation between these factors found in this study suggests that tourists may no longer pay significant attention to COVID-19-related risks. The significant influence of travel intention and monetary promotion indicated a high explanation ratio toward tourists' willingness to support a destination recovery in new normal conditions. Overall, this study has developed and validated the conceptual framework that generates important theoretical and practical contributions.

### 5.1. Theoretical Implications

Despite the enormous influence of the COVID-19 pandemic on the recovery of the worldwide tourism sector [83,84], this study sheds light on the increasing demand for domestic tourism which could be used as an important premise for the revitalization of the tourism industry in the post-pandemic period. The findings in this study reconfirm the critical impact of attitude, perceived behavioral control, and subject norms on tourists' behavioral intention [41,85,86].

This study also makes a unique theoretical contribution to understanding tourists' intention in post-pandemic tourism recovery. Whereas prior research mainly focuses on predicting tourists' behavioral intentions amidst the ongoing pandemic period from 2020 to early 2022 [38,61,68], this study is probably the pioneering research study that investigates tourists' travel behaviors in early 2022 when some countries have decided to reopen their tourism industry to the outside world. While previous studies suggested that destination health risk image might affect tourists' travel decisions [36,87,88], this study provided empirical evidence that this key issue is no longer a major concern influencing their travel intention. This tendency can be traced to some critical reasons. The first reason could stem from the fact that the current Omicron variants of COVID-19 have far lower rates of mortality compared to the initial variants [89,90]. More importantly, the high rate of vaccination coverage, particularly the booster dose policy, has been effectively implemented [91]. As a result, tourists are likely to be more motivated to travel after a long-lasting social isolation or mobility restrictions [92].

Despite the tendency of tourists' increasing travel intention and willingness to support destination recovery after the COVID-19 pandemic, this study provides some evidence that financial measures to adjust tourism products and services play a pivotal role in stimulating tourism demand. Previous studies also highlighted the critical relationship with pricing policy, which might be positively associated with travel intention and supports tourism business revitalization [64,93].

Theoretically, this study also provides an insight into destination crisis management in post-disaster contexts. The findings in this study reconfirm the results from previous studies that the engagement of all stakeholders (e.g., government, tourism operators) might make a key contribution to a tourism destination recovery during and post-pandemic crisis [7,94,95]. Similarly, the empirical evidence from this study reveals the critical impact of social media on tourists' intention to travel and support destination recovery. The contribution of this study emphasizes the significance of using social media as a useful tool to stimulate travel demand in new normal conditions [47,48,96].

### 5.2. Practical Implications

This study has highlighted the positive impact of social media on the travel intention of the domestic market. Therefore, a plethora of social media platforms should be adopted to enhance the destination images in post-pandemic recovery. Importantly, the managers of the tourism destinations and hospitality industry should focus more on tailoring an attractive destination image to boost the domestic tourism demand. At the same time, the word-of-mouth phenomenon remains a popular and trusted non-mainstream platform that has a certain impact on tourists' travel intention and high-quality tourism services and products in the post-pandemic period. The evidence from this study reconfirms the critical role of social media in increasing the domestic demand recovery in a positive way [97]. Thus, rebuilding a tourism destination image by using a wide range of marketing strategies on social media, whether mainstream or non-mainstream media, may effectively contribute to the overall recovery of domestic tourism in the post-pandemic crisis.

However, the domestic tourism recovery may face many potential challenges in the ongoing pandemic crisis. The study suggests that monetary promotion could be used as the most powerful tool to stimulate the increasing demand for domestic tourism in new normal conditions. The findings in this study indicate that the increasing travel intention in post-pandemic recovery is an indispensable demand after lingering lockdowns. According to the

General Statistics of Vietnam [97], the domestic market has witnessed a boom with around 60 million tourist arrivals during the first half of 2022, which nearly doubled the number of tourists in the same period of 2021. More interestingly, the newly registered businesses in the tourism industry have increased by around 27%, which reflects a promising prospect for Vietnam's tourism revival in the post-pandemic crisis. These figures also imply that proper monetary promotion strategies should be developed to accelerate the tourism industry's recovery after the pandemic crisis. In this regard, the tourism industry may stimulate domestic tourist demand by taking proper pricing strategies for all tourism and hospitality services. By doing so, the vast majority of the potential tourists may afford to travel more in the post-pandemic period; thereby the tourism businesses could be gradually revitalized. However, this requires the involvement of different key stakeholders in which both central and local authorities are expected to play a pivotal role in easing the financial pressure that the tourism enterprises are facing. Da et al. [7] found that the domestic tourism system could not rebound without financial packages from the Vietnamese central government. Financial measures to support local enterprises during the pandemic impact have contributed to the tourism business recovery [81,98]. As such, this suggests the central government needs to launch more practical financial support packages to revive the tourism-and-hospitality enterprises and foster the domestic tourism demand to save the post-crisis tourism industry. Despite the necessity of the monetary promotion measures, it may be complicated and challenging for the governments to determine the exact degree to which tourism sectors, genres of businesses, and tourism attractions may be vulnerable to the pandemic and how to ensure competitiveness among the enterprises and destinations. Therefore, such policy involvement may be implemented in the short term.

In addition, the destination health risk image is no longer a big concern for tourists because of the fact that onsite epidemic control has been better managed and health-related risks have been relieved recently. However, the destination managers and tourism operators should continue implementing the epidemic regulations and sustainable practices to be more proactive to the potential future threats of new variants of the coronavirus. To ensure the safety of tourists, the tourism industry may continue advancing technological applications to avoid human contact while retaining high-quality services. Such adaptive transformations should be empowered to improve tourists' on-site experience satisfaction, which is essential for rebuilding an attractive tourism destination image in post-pandemic recovery.

Due to the restricted recovery of international tourism in early 2022 [8], domestic tourism continues to play a decisive role in the revival of the global tourism industry. Therefore, there should be relevant measures to foster domestic tourism more sustainably [99]. This study suggests that tourism destination managers and tourism enterprises should endeavor to maintain on-site tourism experience quality. This continues fostering tourists' revisitation and willingness to support a destination recovery in both the short and long term.

### 5.3. Research Limitations and Future Studies

Although this study has attempted to minimize the limitations, there are a handful of difficulties that should be considered for future research. Notably, the data collection by using the online survey could be replaced by direct questionnaire delivery in new normal conditions when COVID-19 restrictions are unleashed. In addition, more and more international destinations may consider reopening their tourism industry in the post-pandemic crisis as the opportunity to approach international tourists will be available. As such, future studies may consider expanding the research sample by recruiting international tourists. Likewise, potential research may investigate other factors that influence tourists' intention to travel and destination recovery support in other contexts. This helps to provide an insight into the behavioral intention and determinants of destination recovery suitable for different tourism destinations. Ultimately, prospective studies may examine the research phenomenon from other perspectives (e.g., tour operators, destination managers) to better

understand the other key stakeholders; thereby, there will be holistic measures for tourism destination recovery in new normal conditions.

**Author Contributions:** Data curation, D.V.H. and Q.D.P.; Formal analysis, L.H.D. and K.Q.D.; Funding acquisition, Q.D.P. and T.T.N.; Investigation, T.T.N. and T.T.T.; Methodology, L.H.D. and T.T.T.; Project administration, Q.D.P.; Resources, Q.D.P.; Supervision, T.T.N.; Validation, D.V.H.; Writing—original draft, L.H.D.; Writing—review & editing, K.Q.D. All authors have read and agreed to the published version of the manuscript.

**Funding:** This research received no external funding.

**Institutional Review Board Statement:** Ethical review and approval were waived for this study since no (sensitive) personal data and information was processed.

**Informed Consent Statement:** Informed consent was obtained from all subjects involved in the study.

**Conflicts of Interest:** The authors declare no conflict of interest.

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
