# Peer review of "Understanding Tourists’ Behavioral Intention and Destination Support in Post-pandemic Recovery: The Case of the Vietnamese Domestic Market"

_sustainability, doi:10.3390/su14169969_

Round 1

Reviewer 1 Report

Thank you for an interesting approach and valuable findings. The conducted research is worth continuation in future tourist seasons. Also comparative analysis with other markets would be appreciated. If the vast majority of participants were Vietnamese (domestic market), in my opinion, it should be reflected in the title of the article. Some minor corrections are needed (indentation sizes, missing words before table 1.) 

Author Response

Dear Reviewer 1

We would like to express our sincere appreciation for your time and critical comments on our paper. We have carefully addressed all the comments as you can see below.

Point 1: Thank you for an interesting approach and valuable findings. The conducted research is worth continuation in future tourist seasons. Also, comparative analysis with other markets would be appreciated. If the vast majority of participants were Vietnamese (domestic market), in my opinion, it should be reflected in the title of the article

Response 1: Yes, I agree with your suggestion since most of the respondents recruited in this study were Vietnamese tourists. As such, I agree that the word “domestic” could be added to the paper’s title as a great choice better reflect the domestic market’s behavioral intentions in the post-pandemic. Please see the new title

Point 2: Some minor corrections are needed (indentation sizes, missing words before table 1.) 

Response 2: Yes, missing words before Table 1 were removed while the indentation sizes were also addressed. Please see section 4.1

Many thanks again.

Kind regards

Authors

Reviewer 2 Report

reviewer's comments in the attached file

Author Response

Dear reviewer 2

We would like to take this opportunity to express our sincere appreciation for your time and critical comments on our paper. We have carefully clarified all the comments as you can see below.

Point 1: It is only in line 60 (... the tourism industry in Vietnam has hardly hit by the global health crisis ...) that the reader can conclude that the research was conducted on a sample of the inhabitants of Vietnam. This aspect needs to be more emphasized, the more so as the structure of tourism in Vietnam is different from that of, for example, European countries

 Response 1: Yes. We acknowledge that this study recruited a majority of Vietnamese respondents while a limited number of foreign respondents in Vietnam also took part in this survey. In this sense, this study mostly reflects the domestic tourists’ travel intentions when Vietnam entirely re-opened both domestic and international tourism in March 2022. We have clarified this in the Methodology section, 3.2, the first paragraph.

Point 2: In lines 66-73 the authors cite the statistics for 2022, comparing them to the year 2021. It is logical that better indicators were obtained, e.g. the number of arrivals. however, it would be more important to show these figures in relation to the pre-pandemic period in order to get an idea of how the tourism restart is progressing.

Response 2:  Yes, we agree. We have added some key figures to reflect the recovery of the domestic market in pre-and post-pandemic crisis. Please see Section 5.1, paragraphs 1 and 2.

Point 3: It is a pity that the authors do not attempt to define "safe tourism destinations".

Response 3: Yes, we agree. “safe tourism destinations” in this study refers to green zones- the destinations with few or no COVID-19 cases so the tourism activities were encouraged by the Vietnamese government. We have added a few lines to clarify this concept.  Please see section 2.2, paragraph 3.

Point 4: lines 74-75 the authors write ... "little has been known about to what extent tourists are willing to travel in post-pandemic and support destination recovery". This is not a fully true thesis. In addition to various contributory studies, it is worth referring to the comprehensive and systematic 12 waves (2020-2022) of European Travel Commission research under the title “Monitoring sentiment for domestic and intra-European travel”. these are the only updated data in the world based on research in various European countries. The results of these studies answer a number of research questions posed by the authors

Response 4:  Yes, your point is convincing. It could be better if we understate the issue by restating the statement. We have revised it accordingly. Please see section 1, last paragraph.

Point 5: Considerations on the factors of the socio-economic environment and indicators of tourism recovery are complex and ambiguous. This is a significant difficulty in assessing the stage at which tourism is located in individual countries. And it is from this point of view that it seems that the scope adopted by the authors is even too wide

Response 5: Your points are reasonable. We acknowledge that measuring the tourism destination recovery in an individual country is complex, if not challenging in the context of the COVID-19 impact. Because of the complexity of the tourism system, we do not aim to measure the recovery of all tourism sectors but to explore factors influencing tourists’ intentions to travel in post-pandemic and willingness to support destination recovery. Your points elicit us to conduct future research to assess the recovery status of a local tourism system across different sectors by using more relevant indicators. The scope of the study indicates the research objectives which involve the population size and characteristics, geographical location, and the time period of data collection. We have clarified the scope of the study by narrowing it down. Hope it is clearer. Please see Section 3.1 (paragraphs 1 and 3). And section 3.2 (paragraph 1).

Point 6: The authors "jump" on various topics in order to concentrate in the research process on the importance of social media. The reviewer does not argue with the adopted concept, although the importance of SM in marketing communication has been known and described for years

Response 6: We agree with your point. We selectively discuss some relevant topics/ aspects related to popular social media in Vietnam. This helps us reflect on how popular social media, whether mainstream or alternative media, may affect tourists’ travel intentions in the post-pandemic. We have tried improving the coherence of the section 2.4.

Point 7: The reviewer understands the hypotheses, although they seem somewhat straightforward - evident

Response 7: We agree with you. Our literature review reflects the critical relationships between the examined variables, but our study also found new findings in the particular context of post-COVID-19. We think your comment on this issue is helpful for our future research. 

Point 8: In section 2.5 (Monetary Promotion), the authors describe the effect of using sales promotion tools. It is worth adding that during the pandemic, many countries practiced subsidies, tax cuts and tourist vouchers to stimulate demand on the one hand and support tourist entrepreneurs on the other. Such a tool is an effect of the tourism policy and not a sales promotion tool introduced by entrepreneurs. These tools of interventionism and even protectionism are understandable during a pandemic, but questionable from the point of view of the competitiveness of the destination. This is an aspect for a separate discussion, but it is worth paying attention to.

Response 8: This is such an interesting comment which elicits us to improve our paper. We decided to modify section 2.5 (Monetary promotion) and the discussion section (5.2).

Thank you so much for your excellent comments.

Kind regards

Authors

Reviewer 3 Report

This research has important implications for understanding tourism recovery in the post-pandemic era. The research objectives of this paper are clear and valuable, the literature review is rich, and the research conclusions are reasonable. I recommend accepting this paper.

Before publishing, I suggest the authors add some factual discussion of post-pandemic tourism recovery in Vietnam, for example, whether the number of tourists has increased, whether there is a monetary promotion, and how social media is discussing tourism.

Author Response

Dear review 3,

Thank you so much for your time and inspiring comments on our paper’s improvement. We have made some revisions as follows:

Point 1: This research has important implications for understanding tourism recovery in the post-pandemic era. The research objectives of this paper are clear and valuable, the literature review is rich, and the research conclusions are reasonable. I recommend accepting this paper. Before publishing, I suggest the authors add some factual discussion of post-pandemic tourism recovery in Vietnam, for example, whether the number of tourists has increased, whether there is a monetary promotion, and how social media is discussing tourism.

Response 1: Yes, we have briefly added some facts to reflect the importance of monetary promotion and social media in the domestic tourism recovery in post-COVID-19 in Vietnam. Please see section 5.2.

Thank you so much again.

Kind regards

Authors